# The Evolution of Arthroscopic Shoulder Surgery: Current Trends and Future Perspectives

**DOI:** 10.3390/jcm14072405

**Published:** 2025-04-01

**Authors:** Gazi Huri, Ion-Andrei Popescu, Vito Gaetano Rinaldi, Giulio Maria Marcheggiani Muccioli

**Affiliations:** 1Aspetar, FIFA Medical Center of Excellence, Doha 29222, Qatar; gazihuri@hacettepe.edu.tr; 2Department of Orthopaedics and Traumatology, Hacettepe University, 06800 Ankara, Türkiye; 3Romanian Shoulder Institute, ORTOPEDICUM-Orthopaedic Surgery & Sports Clinic, 011665 Bucharest, Romania; ionandrei.popescu@yahoo.com; 4II Clinica Ortopedica e Traumatologica, IRCCS Istituto Ortopedico Rizzoli, 40136 Bologna, Italy; vitogaetanorinaldi@gmail.com; 5DIBINEM—Department of Biomedical and Neuromotor Sciences, University of Bologna, 40126 Bologna, Italy

**Keywords:** arthroscopic shoulder surgery, shoulder instability, rotator cuff repair, Bankart repair, Latarjet, bone block procedure, rotator cuff augmentation, dynamic anterior stabilization, shoulder arthroscopy

## Abstract

Arthroscopic shoulder surgery has undergone significant advancements over the past decades, transitioning from a primarily diagnostic tool to a comprehensive therapeutic approach. Technological innovations and refined surgical techniques have expanded the indications for arthroscopy, allowing minimally invasive management of shoulder instability and rotator cuff pathology. **Methods**: This narrative review explores the historical evolution, current trends, and future perspectives in arthroscopic shoulder surgery. **Results**: Key advancements in shoulder instability management include the evolution of the arthroscopic Bankart repair, the introduction of the remplissage technique for Hill–Sachs lesions, and the development of arthroscopic Latarjet procedures. Additionally, novel techniques such as Dynamic Anterior Stabilization (DAS) and bone block procedures have emerged as promising solutions for complex instability cases. In rotator cuff repair, innovations such as the suture-bridge double-row technique, superior capsular reconstruction (SCR), and biological augmentation strategies, including dermal allografts and bioinductive patches, have contributed to improving tendon healing and functional outcomes. The role of biologic augmentation, including biceps tendon autografts and subacromial bursa augmentation, is also gaining traction in enhancing repair durability. **Conclusions**: As arthroscopic techniques continue to evolve, the integration of biologic solutions and patient-specific surgical planning will likely define the future of shoulder surgery. This review provides a comprehensive assessment of current state-of-the-art techniques and discusses their clinical implications, with a focus on optimizing patient outcomes and minimizing surgical failure rates.

## 1. Introduction

The evolution of arthroscopy began with Thomas Edison’s invention of the incandescent bulb in 1879, which led to the creation of the first cystoscope by Leiter and Nitz in 1886. In 1912, Severin Nordentoft introduced the concept of arthroscopy by using an endoscope to examine the knee joint [1,2]. Kenji Takagi’s pioneering work in Japan laid the groundwork for modern arthroscopy, culminating in the development of the first arthroscope, the Watanabe arthroscope, in the late 1930s [3]. The subsequent introduction of fiber optics and the rod lens system by Hopkins in the mid-20th century revolutionized arthroscopy, expanding its use in a wide range of surgical procedures [4].

While knee arthroscopy gained popularity in the 1970s, shoulder arthroscopy was still in its infancy. The first significant advancements in shoulder arthroscopy began in the early 1980s, with Lanny Johnson’s reports on the procedure [5]. Subsequent developments, such as the introduction of various portals for better visualization and access to the shoulder joint, and the adoption of the beach chair position to mitigate complications associated with the lateral position, marked critical milestones in the evolution of the technique [6,7]. In the last years, several open procedures have started to be performed arthroscopically. The introduction of suture anchors and bio-absorbable tacks, and the refinement of triangulation techniques significantly improved the outcomes of arthroscopic surgeries. The progression from basic diagnostic arthroscopy to complex reconstructive minimally invasive procedures underscores the transformative impact of technological innovations and surgical ingenuity on shoulder arthroscopy. In particular, the arthroscopic Bankart repair, often considered the gold standard for anterior shoulder instability, has seen significant advancements. The addition of the remplissage technique, which addresses engaging Hill–Sachs lesions by filling the defect with the infraspinatus tendon, has shown promising results in reducing recurrence rates and improving shoulder stability [8,9]. Another notable advancement is the arthroscopic Bankart repair with a continuous loop, which enhances the suture configuration, providing stronger tissue fixation and potentially reducing the risk of suture loosening or failure over time. This technique has been particularly beneficial in patients with significant capsular laxity or poor tissue quality [10]. The arthroscopic Latarjet procedure represents a further evolution, offering a robust solution for patients with significant glenoid bone loss. Arthroscopic Latarjet has gained traction due to its ability to combine the advantages of open Latarjet with the benefits of minimally invasive arthroscopy [8]. Another emerging technique, the arthroscopic Dynamic Anterior Stabilization (DAS) with the long head of the biceps (LHB), introduces a novel concept in shoulder stabilization. By utilizing the LHB tendon as a dynamic sling, this technique offers continuous, physiologic support to the anterior shoulder, potentially reducing the need for more invasive procedures in select patients [11]. Similarly, bone block procedures have emerged as a reliable surgical option for treating recurrent anterior shoulder instability, particularly in cases with significant glenoid bone loss or failed previous stabilization attempts [12,13,14,15]. These techniques, which involve the transfer of autologous or allogeneic bone grafts to restore glenoid bone stock, aim to enhance joint stability and reduce the risk of recurrent dislocations. The management of rotator cuff (RC) tears, a prevalent cause of shoulder pain and dysfunction, has been revolutionized by emerging arthroscopic techniques. The introduction of the suture-bridge double-row technique and the development of superior capsular reconstruction (SCR) with autologous biceps tendon or patch augmentation have opened new avenues for effective tendon repair and joint preservation [16,17,18,19]. Additionally, the introduction of the subacromial balloon has provided a less invasive option for managing massive rotator cuff tears [20]. These techniques aim to enhance tendon healing, restore shoulder biomechanics, and improve patient outcomes, particularly in cases of massive or irreparable rotator cuff tears.

This review is structured in three parts, each dedicated to exploring these innovative techniques in detail. The first part will focus on emerging techniques in arthroscopic shoulder instability repair, while the second part will delve into the advancements in arthroscopic RC repair. The final section will discuss the clinical implications of these techniques and potential future directions in arthroscopic shoulder surgery. Through this comprehensive review, we aim to provide a critical assessment of the current state of the art and highlight the future potential of these emerging techniques in improving patient care.

## 2. Shoulder Instability

The shoulder joint has the greatest range of motion in the body but at the cost of an increased risk of instability [21]. Damage or deficiency of the shoulder’s stabilizing structures, whether due to trauma or genetic conditions, disrupts the joint’s balance and predisposes it to luxation. Anterior instability is the most common form, significantly impacting quality of life and, in some cases, completely impairing the professional careers of athletes. Over the years, numerous surgical techniques have been developed to address shoulder instability, each with distinct advantages and limitations [22]. The ideal arthroscopic approach for glenohumeral anteroinferior instability is still an ongoing debate. High failure, recurrence, and overall complication rates after surgery have led to the development of techniques aimed at addressing both soft tissue and bony defects, as well as their interdependence. Initially, a 25% glenoid bone loss was considered the threshold for selecting between soft tissue procedures and bony augmentation techniques [23]. Further research has lowered the value to 20% [24]. Contemporary decision making takes into account not only glenoid bone loss but also its relationship with humeral head bone loss. Studies indicate that glenoid bone loss as low as 13.5% may compromise postoperative outcomes if the Hill–Sachs lesion is not adequately evaluated and addressed [25]. Current surgical options range from traditional Bankart repair to the addition of a remplissage procedure or coracoid transfer, as well as other bone grafting methods to reinforce the anterior glenoid rim. However, the management of cases involving subcritical glenoid bone loss remains a clinical challenge requiring further investigation and refinement [26].

### 2.1. Bankart Repair

The open repair of the anterior capsulo-ligamentous-labral complex, commonly known as the Bankart procedure, was first described by Sir Arthur Sydney Blundell Bankart in 1923. The arthroscopic modification of this procedure was introduced in 1993 and has become the gold standard for the surgical management of anterior shoulder instability [27]. The arthroscopic Bankart repair is a surgical technique designed to restore normal shoulder anatomy by reattaching the labrum–capsule–ligament complex to its original osseous attachment, mirroring the principles of the traditional open procedure. Contemporary fixation methods commonly employ sutures and anchors made from materials such as polyether ether ketone (PEEK), titanium, or advanced all-suture constructs, with configurations that may be knotted or knotless [28]. Although surgical practice often emphasizes simplicity and precision, merely reattaching the torn labrum to the glenoid does not inherently ensure procedural success. Evidence from the literature highlights significant complications, risks, and failure rates associated with Bankart repair. Recurrence of shoulder instability following the procedure ranges from 3% to over 30%, particularly in cases involving bone loss, underscoring the need for careful patient selection and comprehensive surgical planning [29]. The causes of recurrent instability or failure following Bankart repair are multifactorial, involving numerous variables. These include joint laxity, physiological tissue elasticity, healing, patient age, level and type of sports activity, trauma mechanism, timing of the initial injury, recurrence versus first-time dislocation, bone loss on the humeral and/or glenoid side, comorbid conditions, and associated lesions such as rotator cuff tears or posteriorly extended capsulo-labral defects [30]. Therefore, various arthroscopic techniques emerged: arthroscopic remplissage [31], arthroscopic Bankart repair with a continuous loop [32], the arthroscopic Latarjet procedure [8], arthroscopic Dynamic Anterior Stabilization (DAS) with the long head of the biceps (LHB) [11], and arthroscopic Latarjet with screws/buttons/suture cerclage [33].

### 2.2. The Arthroscopic Remplissage

The term “remplissage” is derived from the French word meaning “to fill in”. This surgical technique aims to address the posterior Hill–Sachs defect by utilizing the available infraspinatus capsule and muscle or tendon in a myo-tenodesis fashion. The objective is to reduce the defect’s range of motion relative to the glenoid surface, thereby preventing engagement and subsequent anterior dislocation. The first arthroscopic remplissage procedure was described by Purchase et al. in 2008 [31], and it has since undergone numerous modifications and adaptations. The author’s preferred technique is the parachute approach due to its simplicity and reproducibility, and the lack of need for specialized instruments beyond standard rotator cuff repair instrumentation and implants. The arthroscope is placed into the posterior portal and flipped posteriorly above the humeral head to visualize the defect and the infraspinatus insertion. The assistant maintains slight anterior subluxation of the humeral head by performing an anterior drawer maneuver, thereby expanding the working space around the posterior aspect of the humeral head. Temporary elevation of water pressure may be employed at this stage to further enhance the working space, with care taken to maintain a stable arthroscopic position. A needle is introduced percutaneously into the middle of the defect and a rotator cuff suture anchor of choice is introduced into the marked position. In our opinion, the priority is securing robust soft tissue capture rather than precisely positioning the anchor at the cartilage margin or fully covering the defect. The sutures are passed in a mattress fashion also in a transtendinous way using piercing suture passing instruments like Clever Hooks or Rhinos Sutures. Once all sutures have been passed through the infraspinatus tendon capsule, they can be tied blindly at the end of the procedure, but only after addressing the anterior shoulder repair. During labral repair, tension on the remplissage sutures can be applied to produce a posterior drawer or subluxation of the humeral head, thereby creating additional working space in the anterior shoulder region.

### 2.3. Arthroscopic Subscapular Augmentation

Arthroscopic subscapular augmentation is an advanced surgical technique designed to address shoulder instability, particularly in cases of anterior glenoid bone deficiency with a glenoid bone loss up to 20–25% and contraindications for bone block procedures. This technique involves reinforcing the anterior capsule using the subscapularis tendon or associated tissue structures. It provides an alternative or adjunct to more traditional approaches, such as the Latarjet procedure or open capsular repairs [34]. The arthroscopic subscapular augmentation procedure typically involves several key steps starting with patient positioning, standard arthroscopic portals placement, and a diagnostic arthroscopy to assess the extent of capsular, labral, and bony pathology. The anterior capsule and subscapularis tendon are visualized and mobilized as needed. Using suture anchors, the subscapularis tendon or capsule is securely fixed to the glenoid rim, creating a robust reinforcement of the anterior shoulder [35]. The effectiveness of subscapular augmentation lies in its ability to restore anterior stability by enhancing the dynamic and static restraints of the glenohumeral joint. By reinforcing the anterior capsule, this procedure creates a buttress effect that prevents excessive anterior translation of the humeral head. Additionally, it preserves native anatomy compared to bone block procedures, which may alter glenoid morphology [36]. Emerging clinical evidence suggests favorable outcomes for arthroscopic subscapular augmentation. Key findings from recent studies include improved stability, enhanced functional outcomes, as measured by validated scoring systems such as the Constant–Murley and Rowe scores, and high levels of patient-reported satisfaction and return to pre-injury activity level [37]. Moreover, lower complication rates compared to open procedures and bone block techniques, including reduced risk of neurovascular injury and hardware complications, are to be noted [38,39]. However, limitations exist. The long-term durability of the repair is still under investigation, and the procedure may not be suitable for patients with severe glenoid bone loss or complex bony defects.

### 2.4. Dynamic Anterior Stabilization

Dynamic Anterior Stabilization (DAS) is an innovative surgical approach developed to address anterior shoulder instability, particularly in patients with subcritical glenoid bone loss where traditional soft tissue repairs may be insufficient, and more invasive bony procedures like the Latarjet may be excessive [40]. The DAS procedure involves the arthroscopic transfer of the LHB tendon to the anterior glenoid through a split in the subscapularis tendon. This configuration aims to provide a dynamic restraint against anterior translation of the humeral head, thereby reducing the risk of dislocation [41]. Biomechanically, the transposed LHB tendon acts as a sling at the anterior aspect of the shoulder joint. This dynamic support is particularly beneficial during shoulder abduction and external rotation, positions commonly associated with anterior dislocation events. The dynamic nature of this stabilization contrasts with static procedures, potentially preserving a greater range of motion while providing necessary stability [42,43]. Early clinical studies have reported promising outcomes with DAS. Patients undergoing this procedure have demonstrated low recurrence rates of shoulder instability and satisfactory functional recovery, with further application in skeletally immature patients [44]. Dynamic Anterior Stabilization represents a promising addition to the surgical options for managing anterior shoulder instability. By leveraging the dynamic properties of the LHB tendon, this technique offers a balance between stability and mobility, potentially benefiting patients who are not ideal candidates for more invasive procedures. Ongoing studies and longer-term follow-ups will be crucial in fully elucidating the role of DAS in shoulder instability management.

### 2.5. Between Glenohumeral Ligaments and Subscapularis Tendon (BLS) Technique

An arthroscopic extracapsular stabilization method was developed to address anterior shoulder instability, particularly in patients without significant bone loss (typically less than 10%). This procedure aims to restore the damaged anterior soft tissue structures by reattaching the anterior labrum and capsule to their original footprint [45]. The BLS technique is performed arthroscopically with the patient in the beach chair position under general anesthesia. The procedure involves augmenting the damaged anterior wall soft tissues by incorporating the anterior capsule, the glenohumeral ligaments, and the glenoid labrum in a single construct attached with the help of an anchor far from the articular surface at the 4 o’clock position [45,46]. Early reports suggest that the BLS technique is effective in restoring shoulder stability and function without reducing range of motion. Additional procedures are required when there is significant glenoid bone loss [45,46].

### 2.6. Bone Block Procedures

By recurrent shoulder stability or in cases of significant glenoid bone loss (exceeding 20–25%) or/and large, engaging humeral defects, bone block procedures are effective surgical options for managing anterior shoulder instability. The principle is to augment the anterior glenoid surface with bone grafts, preventing the humeral head from engaging and thereby luxate anteriorly [12].

Bone block procedures can be performed through open or arthroscopic techniques. Both the Latarjet and free bone block procedures have demonstrated success in restoring shoulder stability [47]. However, bone block procedures carry potential risks, including graft resorption, hardware complications, and the development of osteoarthritis [12].

### 2.7. Latarjet Procedure

This involves transferring the coracoid process to the anterior glenoid rim, providing a bone block effect and reinforcing the joint capsule. The Latarjet procedure is well established and has demonstrated efficacy in restoring stability [48] through three mechanisms: reshaping the glenoid, the dynamic sling effect of the conjoint tendon, and the restoration of the coracoacromial arch [49]. The procedure can be performed through open or arthroscopic approaches. In the open technique, a deltopectoral incision is made to access the coracoid process, which is osteotomized and transferred to the anterior glenoid. The graft is then fixed using screws. Arthroscopic techniques have also been developed, offering the potential benefits of minimally invasive surgery, though they require advanced surgical expertise. Long-term studies have demonstrated that the Latarjet procedure results in excellent functional outcomes and a high rate of return to sport among athletes [50,51]. While the Latarjet procedure is effective, it is not without potential complications. These may include recurrence, residual pain, graft nonunion, nerve injury, and the progression of instability arthropathy [52].

### 2.8. Free Bone Block Procedures

These utilize autografts (e.g., iliac crest) or allografts to reconstruct the anterior glenoid. Free bone block procedures are considered effective alternatives to the Latarjet, especially in cases where the coracoid process is unsuitable for transfer [47]. The Eden–Hybinette procedure involves the transplantation of a bone graft, typically harvested from the iliac crest (ICBG), to the anterior glenoid rim in an open or arthroscopic manner [53]. The benefits of using ICBG include the use of autologous donor bone, and the lack of additional cost. This technique is effective for reconstructing large glenoid defects, although it comes with the downside of high donor site morbidity and lacks the sling effect seen in the Latarjet procedure [12].

Distal tibia allografts were introduced in the early 2000s [54] and showed good clinical results with minimal graft resorption [55]. However, disadvantages such as a steep learning curve, high costs, and low availability [12] have hindered the widespread adoption of this technique. Another autograft became available in 2014, namely the distal clavicle [56]. High availability, low costs, low donor site morbidity, and the presence of osteochondral substances were not enough to establish this technique in daily practice and no follow-up studies are available in the literature [13,14].

## 3. Rotator Cuff Repair

Despite many scientific studies and clinical experiences, high retear rates continue to be reported in rotator cuff repair [57,58]. Hopefully, considering that most patients with rotator cuff tears (RCTs) over a certain age are asymptomatic, most retear patients are also generally asymptomatic. However, the dramatic increase in the number of rotator cuff repairs in the last two decades has resulted in a large number of retear patients, and this has led shoulder surgeons to find different repair techniques and augmentation methods to improve the strength of cuff repair and clinical scores. This review focuses mainly on current augmentation techniques and the repair techniques that facilitate biologic healing cascades.

### 3.1. Current Repair Techniques

Early efforts in rotator cuff repair focused on optimizing surgical techniques, with particular attention given to the comparison of single-row, double-row, and transosseous-equivalent (TOE) repairs [59,60]. Single-row repair, which uses fewer anchors and involves less complex techniques, has been favored for its simplicity and cost-effectiveness. However, its biomechanical limitations, particularly a reduced tendon-to-bone contact area, have prompted surgeons to explore alternatives like double-row and transosseous repairs. Double-row repair aims to replicate the anatomic footprint of the rotator cuff more effectively by employing medial and lateral anchor points, thereby enhancing biomechanical stability and healing potential. Transosseous repair, once considered the gold standard before the advent of suture anchors, has recently regained popularity due to its ability to eliminate anchor-related complications, such as anchor pullout or loosening, while also reducing costs [60,61,62].

Despite these biomechanical differences, clinical outcomes among these techniques remain a topic of debate. A systematic review by Ponugoti et al. found no significant differences in pain scores, range of motion, or functional outcomes, such as ASES and UCLA scores, when comparing transosseous-equivalent double-row techniques to complex single-row methods. However, TOE double-row repair demonstrated a lower retear rate compared to simple single-row repair, suggesting its superiority in maintaining tendon integrity over time [60].

Similarly, Gu et al. performed a meta-analysis focusing on tear size as a variable. For smaller tears (<3 cm), no significant clinical differences were observed between single-row and double-row repairs. However, for larger tears (≥3 cm), double-row repair provided better functional outcomes, including improved ASES and UCLA scores, and a significantly lower retear rate [61]. These findings show the importance of tear size in guiding the choice of repair technique.

Stenson et al. emphasized the technical advantages of transosseous techniques, particularly their ability to provide more comprehensive footprint coverage and improved tendon–bone contact without a reliance on anchors. This eliminates anchor-related complications while maintaining cost-effectiveness. However, their analysis also confirmed that clinical outcomes, including pain relief and functional improvement, remain comparable across all techniques [62].

Taken together, these studies suggest that while double-row and transosseous techniques may offer biomechanical advantages, particularly for larger tears, the overall clinical outcomes are largely equivalent across repair methods [60,61]. The choice of technique should therefore be tailored to individual patient factors, including tear size, tendon quality, and cost considerations, rather than a universal preference for one method over another.

### 3.2. Superior Capsular Reconstruction

Superior Capsular Reconstruction (SCR) emerged as an innovative solution for massive, irreparable rotator cuff tears, aiming to restore shoulder biomechanics and stability through the use of grafts anchored between the superior glenoid and the greater tuberosity [63]. The technique, initially introduced by Mihata et al., demonstrated its ability to reduce superior humeral head translation and restore shoulder kinematics in biomechanical studies, laying the groundwork for its clinical adoption [64]. Early clinical studies highlighted promising outcomes, particularly in terms of pain relief and functional recovery, in carefully selected patient populations with minimal arthritis and intact subscapularis tendons [65,66].

Subsequent studies, however, have tempered these initial expectations. Recent analyses have highlighted the variability in outcomes, revealing that SCR may not consistently deliver superior results across broader patient populations [19,63,65,67]. For instance, Werthel et al. systematically reviewed clinical data on SCR and noted significant improvements in functional scores such as ASES, Constant, and SSV, alongside better ROM. However, they also emphasized that graft failure rates remain a major concern, particularly with xenografts and thinner dermal allografts [19].

The type of graft plays a pivotal role in determining the success of SCR. Fascia lata autografts, as championed by Mihata et al., offer superior biomechanical properties and better healing potential but are associated with donor site morbidity [65,66,68]. Dermal allografts, which eliminate the need for autograft harvesting, have gained popularity due to reduced surgical complexity and morbidity. However, their variable healing rates, particularly with thinner grafts (<3 mm), have raised concerns about long-term efficacy [63]. Studies have also explored the use of xenografts and synthetic materials, but these options have shown limited success, often due to higher complication rates and poor integration [19,67,69,70].

The inconsistency in clinical outcomes may also be attributed to factors such as tear chronicity, fixation techniques, and patient selection. Chronic tears often present with significant muscle atrophy and fatty infiltration, reducing the likelihood of successful graft integration and functional recovery [63]. Fixation techniques also vary widely across studies, with no standardized approach to optimize graft tensioning and stability [71,72].

In a more recent critique, Lädermann and Rashid questioned the long-term sustainability of SCR, emphasizing the need for high-quality comparative studies to establish its true place in the therapeutic options [67,73]. They pointed out that while SCR can improve short- to mid-term outcomes in selected cases, its effectiveness compared to alternative treatments such as tendon transfers or reverse total shoulder arthroplasty remains unclear [67].

Despite its challenges, SCR continues to evolve, with ongoing research aimed at refining graft materials, surgical techniques, and patient selection criteria. High-quality randomized controlled trials are essential to better define its role in the management of irreparable RCTs and to optimize its outcomes.

### 3.3. Repair Techniques with Biological Stimulation

It has become clear that techniques such as the double-row repair did not actually create miracles [60]. Even the transosseous repair techniques that were once gold standard became popular again, and it became a matter of debate whether expensive anchor systems were really necessary. Successful outcomes of different transosseous repair techniques reminded us once again that the secret of success is tendon–bone healing [62,74]. There are methods suggested in the literature for biological stimuli, regardless of the repair technique used. The most well-known technique is bone marrow stimulation (BMS) with a microfracture in the footprint or lateral to the greater tuberosity. Pulatkan et al. reported decreased retear rates with microfracture-augmented single-row repair compared to only single-row and double-row repair [75]. Two different meta-analyses, including randomized controlled studies, concluded that BMS was successful in decreasing retear rates; however, both meta-analyses failed to show a statistically significant increase in clinical scores of the patients with BMS [76]. Because it is an easy and inexpensive method to mimic the advantage of techniques such as PRP, bone marrow aspirate, and growth factor injections, most authors recommend adding the bone marrow stimulation method to the repair technique.

The idea that the subacromial bursa, which protects the rotator cuff mechanically and surrounds it as a biological envelope, may also aid in healing has been tested in animal and clinical studies [77,78]. Transfer of the subacromial bursa to the repair site was first described by Freislederer et al. [79]. Another study by Guler et al. reported its applicability in partial bursal side tears [80]. Although many studies have been conducted on the biological role of the subacromial bursa in the pathophysiology of rotator cuff tears, there are few clinical studies on its effect in the repair area. Nevertheless, the literature suggests to avoid extensive bursectomy as it may result in more subacromial adhesions [81]. We are of the opinion that further studies are needed to clarify its usefulness as a biological augmentation and it seems more logical not to perform an unnecessary extensive bursectomy to maintain biological stimulation from bursae.

The application of some injectable solutions that accelerate biological healing, such as PRP or various growth factors, to the repair area is also quite popular, but unfortunately, despite many studies, they have not been proven to be clinically effective [82,83]. Since this review focuses on augmentation techniques, it will not go into detail about this subject as it is a very broad and comprehensive topic.

### 3.4. Dermal Allografts

Patch augmentation with dermal allografts has gained recognition as a biologic approach to enhance healing in rotator cuff repairs, particularly for large, massive, or degenerative tears [84]. These allografts act as a scaffold, promoting cellular infiltration and neovascularization, both critical for effective tissue integration and long-term healing [85]. By providing an extracellular collagen matrix, dermal allografts mimic the structure of natural tendons, encouraging the body’s healing mechanisms to regenerate damaged tissue [86]. Multiple studies highlight the efficacy of dermal allografts in reducing retear rates [87].

Arthroscopic techniques using human dermal allografts have demonstrated significant improvements in biomechanical strength and healing outcomes, especially in patients with poor tendon quality [88,89].

Haque et al. evaluated outcomes of open dermal allograft bridging repair for massive irreparable rotator cuff tears. In 22 patients, significant improvements in clinical scores and range of motion were observed at a mean follow-up of 2.8 years. Although 36% of patients experienced retears, the retear size was smaller, and functional improvements were maintained, suggesting this approach can delay the need for reverse shoulder arthroplasty [90]. Recent advancements in arthroscopic techniques have further highlighted the potential of dermal allograft augmentation. The “Canopy” technique, described by Hirahara et al. employs a double-row repair with dermal allograft augmentation, offering enhanced structural support for thinned or compromised rotator cuff tissue while simplifying graft fixation and reducing surgical complexity [91]. Gardner et al. introduced a technique for arthroscopic rotator cuff repair using a cannulated dermal allograft implant, which minimizes surgical time, reduces graft waste, and effectively augments repair at the weakest areas, enhancing structural support [92].

Despite these advantages, challenges remain. Dermal allografts can be costly, and their application requires technical expertise, particularly in arthroscopic procedures [93]. Additionally, long-term data on their durability and effectiveness in varied patient populations are still limited [94].

### 3.5. Bioinductive Patches

Bioinductive patches, such as the REGENETEN^®^ scaffold, represent an innovative approach in the biologic augmentation of rotator cuff repair. These collagen-based scaffolds are designed to enhance tendon healing by stimulating tissue regeneration at the tendon–bone interface [94]. By inducing the formation of new tendon-like tissue, bioinductive patches reinforce the repair site, improve structural integrity, and provide a preventative measure against tear progression. Clinical applications have shown particular efficacy in managing partial-thickness tears and early-stage rotator cuff injuries, offering a proactive solution to mitigate tear progression [95].

In a meta-analysis, Warren et al. highlighted the clinical advantages of bioinductive patch augmentation [96]. Their analysis demonstrated significantly lower retear rates with patch augmentation, with rates as low as 1.1% for partial-thickness repairs and 8.3% for full-thickness repairs, compared to traditional, non-augmented techniques. Additionally, tendon thickness increased markedly postoperatively, correlating with improved functional outcomes, including higher ASES and Constant–Murley scores and reduced patient-reported pain levels outcomes. Thon et al. further emphasized the versatility and efficacy of REGENETEN patches in their review of the current literature, highlighting consistent improvements in tendon healing and clinical outcomes. They reported that the scaffold integrates seamlessly into native tissue, with minimal complications, and is particularly beneficial in patients at higher risk of retear [97].

Kantanavar et al. provided additional evidence of the effectiveness of patch augmentation in patients with large to massive posterosuperior tears. Their retrospective comparative study demonstrated that repair with human dermal allograft patch augmentation resulted in superior functional scores, reduced pain, and significantly lower retear rates compared to single-row repair alone, supporting the broader application of biologic scaffolds in addressing complex tear patterns [95].

Recent studies have reinforced the benefits of bioinductive patches in tendon repair; Camacho-Chacon et al. demonstrated significant increases in tendon thickness and clinical improvements in patients treated with bioinductive patches [98]. Biopsies revealed the formation of tendon-like tissue indistinguishable from native tendon, confirming the patch’s role in promoting biologic integration and healing. Clinical outcomes included marked improvements in functional scores and reduced pain, with MRI imaging showing the healing progress. Bokor et al. demonstrated that bioinductive patches improve tendon healing and reduce retear rates significantly. Patients experienced consistent improvements in functional outcomes, and MRI evaluations demonstrated successful graft integration and increased tendon thickness [99].

These results highlight the potential of bioinductive patch augmentation to not only facilitate tendon healing but also improve long-term functional outcomes. By reducing retear rates and promoting durable repairs, bioinductive patches represent a promising solution for high-risk patients or those with complex tear patterns. As the evidence base continues to grow, bioinductive patches may become an integral component of rotator cuff repair strategies.

### 3.6. Biceps Augmentation

Many biceps augmentation techniques have been described in the literature. Neviaser et al. introduced the first idea of biceps augmentation for irreparable cuff tears [100]. The advantage of using a biceps tendon as a graft lies on its natural source of tenocytes as it is already a tendon itself. There are many techniques available for using the tendon directly (inlay) in the repair area or by using it as a patch (onlay). Although the fact that the biceps tendon is not very wide in shape poses an obstacle to its use as a patch, histological studies showed that smash techniques such as dermal graft preparation did not affect the viability of tenocytes in the biceps tendon [101]. Many techniques were published to prepare the tendon as a patch after the harvest and make it suitable for augmentation [102,103,104]. Other techniques used this tendon with its original shape after tenotomy or without tenotomy even to augment the superior capsular reconstruction [105,106,107].

Biceps augmentation techniques in rotator cuff repair have evolved significantly, offering various ways to maximize the tendon’s biological and mechanical benefits [108]. Some techniques involve reshaping the biceps tendon for onlay (patch) applications, where it acts as a biological scaffold over the tear site [109]. This method can be particularly useful in cases where tendon width is insufficient; techniques such as “smash” or compression procedures flatten the tendon, increasing its surface area and enabling it to cover larger defects [103]. Additionally, arthroscopic methods allow the long head of the biceps tendon (LHBT) to be anchored directly to the rotator cuff footprint, reinforcing the repair and providing stability [107]. The “compressed biceps autograft” is another variation, where the tendon is reshaped and secured using minimal fixation to enhance the healing environment [104].

Further, LHBT has been used to augment superior capsular reconstruction (SCR) in irreparable tears, often without tenotomy to preserve the proximal insertion [110,111]. This technique repositions the tendon to serve as a dynamic stabilizer, reducing superior humeral migration and mimicking the function of the rotator cuff cable [105,111]. Comparatively, biceps autografts have been shown to improve tendon healing in cases of poor-quality tissue, providing additional tenocytes and improving vascularity at the repair site. These techniques, whether inlay, onlay, or adapted for SCR, reflect the versatility of LHBT augmentation in addressing complex rotator cuff tears [112].

The results of biceps augmentation in rotator cuff repair is promising. A recent meta-analysis showed reduced retear rates in patients with biceps augmentation; however, the authors could not find a difference in clinical scores of the patients [113]. Green et al. reported in their systematic review that using long head of biceps augmentation techniques resulted in superior clinical outcomes in large to massive cuff tear patients [114].

### 3.7. Tendon Transfers

Tendon transfers are a surgical solution for massive, irreparable rotator cuff tears that restore shoulder joint stability and function by redirecting muscle forces [115,116]. The primary goal is to re-establish force coupling and rotational strength, which are compromised in massive rotator cuff tears. Ideal candidates for tendon transfer are younger, motivated patients with minimal glenohumeral joint degeneration and significant functional deficits due to irreparable tears [117]. Commonly utilized tendon transfer options include the latissimus dorsi and pectoralis major for specific tear patterns, though each has limitations based on anatomical differences and postoperative outcomes [118,119]. We only focused on a newer technique with lower trapezius transfer in this review.

### 3.8. Lower Trapezius Transfer

Among these options, the lower trapezius (LT) transfer is increasingly favored due to its anatomical alignment and biomechanical similarity to the infraspinatus, making it particularly effective in restoring external rotation in patients with posterosuperior cuff tears [120]. The LT transfer procedure involves attaching the LT to the humerus using an Achilles tendon allograft or semitendinosus graft to bridge the gap [121]. Studies have shown that LT transfer can improve external rotation, reduce pain, and enhance overall shoulder function [122,123]. Additionally, both open and arthroscopically assisted techniques are available, with the latter demonstrating reduced complications and faster recovery [124].

### 3.9. Latissimus Dorsi Transfer

Latissimus dorsi tendon transfer (LDTT) is a well-established option for irreparable posterosuperior rotator cuff tears, particularly in patients with severe functional deficits. Originally described by Gerber et al. [125], LDTT helps restore external rotation and shoulder elevation, though outcomes depend on patient selection and subscapularis integrity. While lower trapezius transfer (LTT) offers better biomechanical alignment, LDTT remains a valuable alternative when LTT is not feasible. Further studies are needed to refine indications and compare long-term outcomes.

## 4. Conclusions

Arthroscopic shoulder surgery has evolved significantly, offering minimally invasive solutions for shoulder instability and rotator cuff pathology. Advancements such as the remplissage technique, arthroscopic Latarjet, and Dynamic Anterior Stabilization have improved the management of instability, while superior capsular reconstruction, dermal allografts, and bioinductive patches have enhanced rotator cuff repair outcomes.

Moreover, while arthroscopic shoulder surgery has advanced significantly, open procedures remain relevant in select cases. Arthroscopic techniques offer reduced morbidity, faster recovery, and lower complication rates, particularly in instability management and rotator cuff repair. However, open approaches may still be preferable in cases requiring extensive bone augmentation (e.g., Latarjet for severe glenoid bone loss) or massive rotator cuff tears with poor tendon quality, where direct visualization and robust fixation are advantageous.

Despite these innovations, challenges remain in optimizing long-term results and reducing failure rates, and economic impact.

In conclusion, arthroscopic shoulder surgery has evolved into a highly sophisticated field, offering innovative solutions for both instability and rotator cuff pathology. Advances such as arthroscopic remplissage, Latarjet, and Dynamic Anterior Stabilization have expanded treatment options for instability, while superior capsular reconstruction, bioinductive patches, and tendon augmentation techniques have improved outcomes in rotator cuff repair. However, challenges remain, including optimizing long-term success rates, minimizing complications, and balancing cost-effectiveness with surgical innovation. A critical comparison with open procedures highlights the benefits of minimally invasive techniques while acknowledging the continued role of open surgery in complex cases. Future research should focus on refining patient selection criteria, enhancing biological integration of grafts, and assessing the long-term cost–benefit ratio of emerging technologies. The integration of biologics and personalized surgical planning will likely shape the next frontier in shoulder surgery.

Future research should focus on refining biologic augmentation, improving graft integration, and personalizing surgical strategies to enhance patient outcomes. The continued integration of technology and biologics will likely shape the next era of arthroscopic shoulder surgery.

## Data Availability

The datasets used and analyzed during the current study are available from the corresponding author.

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
