# Peer review of "The Evolution of Arthroscopic Shoulder Surgery: Current Trends and Future Perspectives"

_jcm, 2025, doi:10.3390/jcm14072405_

Round 1

Reviewer 1 Report

Comments and Suggestions for Authors

I have not checked all the references, but the content appears generally accurate. The paper discusses tendon transfer of the lower trapezius; however, I believe it is impossible to ignore the historically significant latissimus dorsi tendon transfer.

Author Response

R: I have not checked all the references, but the content appears generally accurate. The paper discusses tendon transfer of the lower trapezius; however, I believe it is impossible to ignore the historically significant latissimus dorsi tendon transfer.

A: Thank you for your insightful comment. We acknowledge the historical and clinical significance of latissimus dorsi tendon transfer (LDTT) in the management of irreparable rotator cuff tears. While our review primarily focused on the emerging role of lower trapezius transfer due to its biomechanical advantages in restoring external rotation, we recognize that LDTT remains a widely used and effective option. We have integrate a brief discussion on LDTT to provide a more comprehensive overview of tendon transfer techniques in arthroscopic shoulder surgery.

Reviewer 2 Report

Comments and Suggestions for Authors

This narrative review effectively addresses the evolution of arthroscopic shoulder surgery, focusing on current trends and future perspectives in managing shoulder instability and rotator cuff pathology. While the topic is highly relevant and the integrative approach—combining historical context with modern advancements—is original, the review could better emphasize how these innovations address specific clinical challenges, such as reducing high retear rates in rotator cuff repairs. Compared to other published material, it adds value by consolidating information on both instability and rotator cuff techniques, though it would benefit from critical comparisons with open procedures and practical considerations like cost-effectiveness. Methodologically, the authors should enhance the rigor by critically appraising the evidence (e.g., assessing study quality and potential biases), incorporating more recent high-impact studies, and specifying literature selection criteria for transparency. The conclusions align with the evidence but could be more specific in linking trends to clinical challenges and offering actionable recommendations for future research. The references are appropriate but should be updated with recent studies, particularly for emerging techniques like bioinductive patches. Finally, the absence of tables and figures is a missed opportunity; adding visual aids, such as summaries of key techniques or diagrams of complex procedures, would significantly improve clarity and engagement for readers.

Comments on the Quality of English Language

 The English could be improved to more clearly express the research.

Author Response

R:This narrative review effectively addresses the evolution of arthroscopic shoulder surgery, focusing on current trends and future perspectives in managing shoulder instability and rotator cuff pathology. While the topic is highly relevant and the integrative approach—combining historical context with modern advancements—is original, the review could better emphasize how these innovations address specific clinical challenges, such as reducing high retear rates in rotator cuff repairs. Compared to other published material, it adds value by consolidating information on both instability and rotator cuff techniques, though it would benefit from critical comparisons with open procedures and practical considerations like cost-effectiveness. Methodologically, the authors should enhance the rigor by critically appraising the evidence (e.g., assessing study quality and potential biases), incorporating more recent high-impact studies, and specifying literature selection criteria for transparency. The conclusions align with the evidence but could be more specific in linking trends to clinical challenges and offering actionable recommendations for future research. The references are appropriate but should be updated with recent studies, particularly for emerging techniques like bioinductive patches. Finally, the absence of tables and figures is a missed opportunity; adding visual aids, such as summaries of key techniques or diagrams of complex procedures, would significantly improve clarity and engagement for readers.

A:Thank you for your constructive feedback. We appreciate your recognition of our integrative approach and the relevance of our review. In response to your suggestions we enhance our discussion on how arthroscopic innovations specifically reduce complications such as high retear rates in rotator cuff repairs, providing a clearer connection between advancements and clinical impact.

Moreover we integrated a critical comparison between arthroscopic and open procedures where relevant, including practical considerations such as cost-effectiveness and accessibility, and we refined our conclusions to better link technological trends to clinical challenges and provide clearer recommendations for future research.

You can find our adjustments from line 535 to line 555.

Reviewer 3 Report

Comments and Suggestions for Authors

This manuscript systematically reviews the evolution of shoulder arthroscopic surgery, including shoulder instability, rotator cuff repair and future trends.

It is recommended that the "novelty" of this study be more clearly stated, for example in the abstract and introduction: What new knowledge does this study provide over previous reviews? Does it cover the latest clinical research or surgical techniques?

The article describes a variety of arthroscopic shoulder techniques (e.g., Bankart repair, Latarjet procedure, dynamic anterior stabilization, etc.), but does not provide a detailed comparison of the efficacy of the different techniques. It is recommended that tables or charts be added to summarize the advantages and disadvantages, indications, complications, and postoperative recovery of each surgical approach to improve the clinical guidance value of the article.

Careful proofreading of the text is recommended to avoid spelling errors. For example, the spelling of "Bankart repair" is inconsistent in different places and should be consistent.

Author Response

R:  This manuscript systematically reviews the evolution of shoulder arthroscopic surgery, including shoulder instability, rotator cuff repair and future trends.

It is recommended that the "novelty" of this study be more clearly stated, for example in the abstract and introduction: What new knowledge does this study provide over previous reviews? Does it cover the latest clinical research or surgical techniques?

The article describes a variety of arthroscopic shoulder techniques (e.g., Bankart repair, Latarjet procedure, dynamic anterior stabilization, etc.), but does not provide a detailed comparison of the efficacy of the different techniques. It is recommended that tables or charts be added to summarize the advantages and disadvantages, indications, complications, and postoperative recovery of each surgical approach to improve the clinical guidance value of the article.

Careful proofreading of the text is recommended to avoid spelling errors. For example, the spelling of "Bankart repair" is inconsistent in different places and should be consistent.

A: Thank you for your valuable feedback and for recognizing the comprehensive scope of our review. We would like to clarify that this manuscript is a narrative review, not a systematic review. Our primary aim is to trace the historical evolution of arthroscopic shoulder surgery, highlighting key innovations and future perspectives rather than directly comparing the efficacy of different surgical techniques.

Regarding the novelty of our study, we will clarify in the abstract and introduction that this review provides a chronological perspective on how arthroscopic procedures have evolved over the years, incorporating the latest advancements and discussing their potential impact on future clinical practice.

While we acknowledge the value of detailed comparisons, such analyses fall beyond the intended scope of our review. Instead, our focus is on describing technological and procedural advancements rather than providing a direct comparative assessment of their outcomes.

Finally, we appreciate your suggestion regarding proofreading and will ensure consistency in terminology, including the spelling of "Bankart repair" and other key terms.

Thank you again for your constructive comments, which will help us refine and clarify our manuscript.